# Thermal Comfort Conditions and Mortality in Brazil

**DOI:** 10.3390/ijerph21091248

**Published:** 2024-09-20

**Authors:** Weeberb J. Requia, Reizane Maria Damasceno da Silva, Leonardo Hoinaski, Heresh Amini

**Affiliations:** 1Center for Environment and Public Health Studies, School of Public Policy and Government, Fundação Getulio Vargas, Brasilia 72125590, Brazil; reizanemaria1@gmail.com; 2Sanitary and Enviromental Engineering Department, Universidade Federal de Santa Catarina, Florianópolis 88040600, Brazil; leonardo.hoinaski@ufsc.br; 3Icahn School of Medicine at Mount Sinai, New York, NY 10029, USA; heresh.amini@mssm.edu

**Keywords:** thermal stress, mortality, respiratory diseases, cardiovascular diseases

## Abstract

Conventional temperature-based approaches often overlook the intricate nature of thermal stress experienced by individuals. To address this limitation, climatologists have developed thermal indices—composite measures designed to reflect the complex interaction of meteorological factors influencing human perception of temperature. Our study focuses on Brazil, estimating the association between thermal comfort conditions and mortality related to respiratory and circulatory diseases. We examined four distinct thermal indices: the discomfort index (DI), net effective temperature (NET), humidex (H), and heat index (HI). Analyzing a comprehensive dataset of 2,872,084 deaths from 2003 to 2017, we found significant variation in relative risk (RR) based on health outcomes, exposure lag, percentile of exposure, sex/age groups, and specific thermal indices. For example, under high exposure conditions (99th percentile), we observed that the shorter lags (3, 5, 7, and 10) had the most robust effects on all-cause mortality. For example, under lag 3, the pooled national results for the overall population (all ages and sexes) indicate an increased risk of all-cause mortality, with an RR of 1.17 (95% CI: 1.13; 1.122) for DI, 1.15 (95% CI: 1.12; 1.17) for H, 1.15 (95% CI: 1.09; 1.21) for HI, and 1.18 (95% CI: 1.13; 1.22) for NET. At low exposure levels (1st percentile), all four distinct thermal indices were linked to an increase in all-cause mortality across most sex and age subgroups. Specifically, for lag 20, we observed an estimated RR of 1.19 (95% CI: 1.14; 1.23) for DI, 1.12 (95% CI: 1.08; 1.16) for H, 1.17 (95% CI: 1.12; 1.22) for HI, and 1.18 (95% CI: 1.14; 1.23) for NET. These findings have important implications for policymakers, guiding the development of measures to minimize climate change’s impact on public health in Brazil.

## 1. Introduction

The connection between climate and human health has increasingly captured the focus of environmental epidemiologists [1,2,3,4,5,6,7]. Traditionally, ambient temperature has been the primary measure for evaluating climate’s impact on health. However, temperature alone provides a limited perspective. Thermal comfort, encompassing factors such as humidity, wind speed, and radiant heat, offers a more comprehensive understanding of how people perceive and react to extreme climate conditions, including suboptimal temperatures [8]. Research has shown that these thermal comfort metrics reveal the complex interactions between climate variables and human physiology [9,10,11].

Despite significant advances in understanding the relationship between climate and health, several critical research gaps remain. First, most existing studies have been concentrated in temperate regions [10,12,13] with far fewer investigations conducted in tropical areas like Brazil, where unique climate patterns and socioeconomic conditions could lead to different health impacts. Tropical regions experience higher humidity and more frequent extreme heat events, which could exacerbate health risks, yet these factors are underrepresented in current research. Additionally, while some studies have explored the relationship between temperature and health, [10,12,13] there is a lack of comprehensive research that examines the impact of a broader range of thermal comfort indices, such as the net effective temperature (NET) and heat index (HI), which integrate multiple meteorological variables. These indices offer a more nuanced understanding of thermal comfort but remain underutilized in epidemiological studies, limiting our ability to assess their true effects on health outcomes such as respiratory and cardiovascular diseases.

Another underexplored area is the effect of lag periods between exposure to thermal discomfort and the manifestation of health outcomes. The time delay, or lag, between exposure to extreme thermal conditions and adverse health effects is poorly understood, particularly in terms of how different lag periods (short-term vs. long-term) may influence mortality and morbidity risks. This gap is crucial for developing timely public health interventions. Furthermore, many existing studies overlook how demographic factors such as age, gender, and socioeconomic status modulate the relationship between thermal comfort and health [14,15,16,17]. Vulnerable populations, including the elderly, children, and low-income groups, may face heightened risks from thermal stress due to pre-existing health conditions, inadequate access to cooling, and limited healthcare resources [18,19]. Addressing these gaps is essential for creating targeted climate adaptation strategies that promote health equity in regions like Brazil, where socioeconomic disparities are pronounced.

This study addresses these gaps by examining the association between thermal comfort conditions and mortality for respiratory and circulatory diseases in Brazil. By focusing on four distinct thermal indices and considering various lag periods and demographic factors, this research provides a comprehensive analysis of thermal comfort in a diverse and expansive country. Brazil’s vast geographical expanse and significant socioeconomic disparities make it an ideal setting for exploring these dynamics in detail.

## 2. Methods

### 2.1. Mortality Data

We obtained mortality data from the Brazilian Ministry of Health, covering the period from 1 January 2003 to 31 December 2017. This dataset includes records of 2,872,084 deaths across Brazil’s 5572 municipalities. In Brazil, municipalities represent the smallest administrative divisions, which are further grouped into five major regions: North, Northeast, Midwest, Southeast, and South. The spatial layout of these municipalities and regions is presented in Appendix A.

For this study, we narrowed the dataset to include only deaths attributed to respiratory diseases (ICD-10 codes J00-J99) and circulatory diseases (ICD-10 codes I00-I99), yielding a subset of 1,071,090 records. These mortality records were then stratified by sex, age group (0–14, 15–45, 46–65, and >65 years), geographic region (North, Northeast, Midwest, Southeast, and South), and cause of death (respiratory or circulatory diseases).

Although we did not perform a formal power calculation, the scale and comprehensiveness of the mortality data in this study inherently provide a strong basis for statistical power. The data were sourced from the Brazilian Ministry of Health’s Mortality Information System (SIM), the official and national repository of mortality records in Brazil. This system covers the entire country, ensuring a high degree of completeness and reliability in the data collection process. With over 1,071,090 deaths from circulatory and respiratory diseases included in the analysis, spanning 5572 municipalities across 15 years, the size and national coverage of the dataset ensure a high likelihood of detecting meaningful associations. The large sample size, combined with a nationwide representation, increases the precision of our estimates and reduces the risk of Type II errors (failing to detect an effect when one exists).

### 2.2. Weather Data

Meteorological data were obtained from the Copernicus ERA5-Land (https://cds.climate.copernicus.eu/cdsapp#!/dataset, accessed on 11 December 2023). The dataset encompasses daily measurements of surface temperature (mean, in °C), wind speed (m/s), and humidity (in %). The dataset features a spatial resolution of 10 km. To align the data with the administrative boundaries of Brazilian municipalities, we used a zonal mean approach, which calculates averages based on the geographic coordinates of each municipality.

### 2.3. Thermal Indices

To assess outdoor human comfort, we employed four thermal indices derived from linear equations. A brief description of each index is provided below along with a description of the strengths and limitations of each index:Discomfort index (DI): This index merges average air temperature and relative humidity to gauge outdoor comfort levels. Values under 15 and above 26.5 signal discomfort, while those between 15 and 20 are considered comfortable, and values from 20 to 26.5 indicate slight discomfort. The DI combines air temperature and relative humidity to assess thermal comfort. One of its strengths lies in its simplicity and its long history of use in public health research. However, DI does not account for wind speed, which can be a critical factor in moderating heat stress, especially in warmer regions like Brazil. This limitation may reduce its ability to capture thermal discomfort in areas with high winds, where wind chill could affect perceived temperature.Net effective temperature (NET): This index evaluates outdoor thermal comfort by factoring in air temperature, relative humidity, and wind speed. A NET value between 17 and 21 denotes comfort, with specific ranges indicating mild discomfort in colder conditions and discomfort in hotter conditions. Its ability to factor in wind speed makes it more accurate in representing outdoor thermal comfort in various climatic conditions, especially in tropical regions where wind can mitigate heat stress. However, NET is more complex to calculate, and its interpretation may be less intuitive than simpler indices like DI or HI, potentially limiting its broader use in public health studies.Humidex (H): By combining air temperature and vapor pressure, the humidex provides a measure of thermal discomfort. Values below 29 suggest no discomfort, those between 30 and 39 imply mild discomfort, and values exceeding 45 points to hazardous discomfort levels. The humidex is well-suited to assess heat stress in regions with high humidity, as it reflects the combined impact of heat and moisture on human comfort. However, like DI, humidex does not consider wind speed, which may lead to an incomplete assessment of thermal stress in windy environments. Additionally, humidex values can sometimes overestimate discomfort in regions with extremely high humidity levels, which may exaggerate the associated health risks.Heat index (HI): This index, derived from air temperature and relative humidity, indicates levels of thermal discomfort. Values under 27 are deemed comfortable, while those over 32 are classified as dangerous. One of HI’s strengths is its familiarity and extensive use in heat-health warning systems, which makes it practical for public health applications. However, similar to DI and the humidex, HI does not include wind speed, which may limit its applicability in windy or cooler regions where wind plays a significant role in moderating thermal discomfort.

For comprehensive information on the formulation and interpretation of each index, please refer to the Appendix A. Overall, the inclusion of multiple indices in this study allows us to capture different aspects of thermal stress across a range of climatic conditions. While indices like NET offer a more comprehensive measure by incorporating wind speed, simpler indices such as DI, humidex, and HI are easier to calculate and interpret. Each index has strengths in different settings, but limitations arise when key meteorological factors—such as wind speed—are excluded, potentially leading to under- or overestimation of thermal stress in specific environments. By using a combination of these indices, we aim to provide a more nuanced understanding of the relationship between thermal comfort and health outcomes in Brazil’s diverse climatic regions.

### 2.4. Statistical Analysis

Our investigation employed an extended two-stage design, integrating a case time series methodology (referencing Gasparrini et al., 2022), [20] to explore the relationship between each thermal index (with detailed models outlined in Section 2.3) and daily mortality from circulatory and respiratory diseases in Brazil spanning from 2003 to 2017. Initially, we conducted a regional assessment of the Brazilian region, utilizing a series at the municipality level. Subsequently, we estimated national effects. Below, we outline the statistical analysis for each stage.

In the initial stage, we utilized a distributed lag non-linear modeling (DLNM) framework proposed by Gasparrini et al. (2010) [21]. Our goal was to create a cross-basis function for each thermal index. This function captures the relationship with each index, accommodating non-linearity and temporal dependencies through the lag component.

For each thermal index, we defined a cross-basis function using natural cubic splines. Specifically, we placed three internal knots at the 10th, 75th, and 90th percentiles. These knots allow us to depict non-linearities in exposure-response relationships effectively. The choice of these parameters was informed by previous studies that explored temperature and health effects [22,23].

To investigate the lagged effect of each thermal index, we considered various lag periods: 3, 5, 7, 10, 15, and 20 days. The cross-basis function incorporated a list of municipalities, defining groups of observations. By exploring different lag periods, we aimed to capture potential delayed effects and gain a comprehensive understanding of the temporal dynamics between thermal indices and mortality. Finally, we selected all parameters for the cross-basis function using the Akaike Information Criterion (AIC).

We integrated the cross-basis functions into a generalized conditional quasi-Poisson regression model. In this model, the exposure was represented by the cross-basis derived from each thermal index. To account for temporal trends and seasonality at the municipality level, we defined a stratum. Time-stratified sampling was employed to create these strata, categorizing them based on the day of the week, month, calendar year, and municipality within the time series.

For each thermal index, we developed individual models and conducted analyses stratified by various factors, including sex, age group (0–14, 15–45, 46–65, and >65 years old), health outcome (all-cause mortality, respiratory mortality, and circulatory mortality), and Brazilian regions (North, Northeast, Midwest, Southeast, and South).

To quantify associations, we estimated specific relative risks (RRs) and their corresponding 95% confidence intervals (CIs). These RRs were calculated over different lag periods (3, 5, 7, 10, 15, and 20 days), capturing the overall cumulative effect. Additionally, we examined extreme percentiles (1st and 99th) to understand associations with both low and high thermal indices. Specifically, for the humidex (H) and heat index (HI), we captured extremely comfortable (1st percentile) and uncomfortable (99th percentile) thermal sensations; while for the discomfort index (DI), we focused on extremely uncomfortable thermal sensations at the 1st and 99th percentiles.

In the second stage, we conducted a meta-analysis to calculate RRs at the national level. To address intra- and inter-region variability, we incorporated random effects for municipalities within regions. Additionally, we applied fixed effects in the meta-regression for regional-scale predictors, including total population, total number of deaths, and gross domestic product (GDP). These predictors were chosen based on previous studies exploring temperature variations [24,25]. We assessed heterogeneity using the I-squared (I2) statistic, considering homogeneity for *p*-values > 0.05 and/or I2 < 50%.

All statistical analyses were performed in R (version 4.0.2) using the following packages: “dlnm” (distributed lag non-linear models) for the distributed lag modeling framework, “gnm” (generalized nonlinear models) for the generalized conditional quasi-Poisson regression model, and “mixmeta” for the meta-analysis.

## 3. Results

Our study analyzed a dataset of 2,872,084 recorded deaths in Brazil from 2003 to 2017 obtained from the Brazilian Ministry of Health. Notably, respiratory diseases contributed to approximately 10% of total deaths, while circulatory diseases accounted for 28%. Male deaths constituted a higher percentage: 58% in all-cause mortality, 54% in respiratory mortality, and 52% in circulatory mortality. Additionally, the majority of individuals affected were aged over 65 years, comprising 69% in respiratory mortality and 66% in circulatory mortality (Table 1).

Table 2 shows the descriptive statistics detailing each thermal index alongside meteorological parameters, including temperature, humidity, and wind speed. Spanning from 2008 to 2018, the average ambient temperature recorded was 23.59 °C, with a standard deviation of 3.79 °C, ranging from 1.25 °C to 34.92 °C. Relative humidity averaged 77.76%, with a standard deviation of 13.64%, while wind speed had an average of 2.99m/s, with a standard deviation of 1.25 m/s. The thermal indices DI, NET, H, and HI had average values of 22.41 (standard deviation = 3.24), 18.26 (standard deviation = 4.59), 30.74 (standard deviation = 6.27), and 24.59 (standard deviation = 4.80), respectively. Figure 1 provides a visual representation of the spatiotemporal distribution of each thermal index across Brazil during the study period.

Our results suggest considerable variation depending on the health outcome (all-cause, respiratory, and circulatory mortality), exposure lag (3, 5, 7, 10, 15, and 20 days), percentile of exposure (low, 1st percentile; high, 99th percentile), sex/age groups, and the thermal indices (DI, H, HI, and NET). For example, at low exposure levels (1st percentile), all four distinct thermal indices were linked to an increase in all-cause mortality across most sex and age subgroups, although a few associations were statistically insignificant (e.g., people aged 0–14 years old). This trend remained largely consistent across various exposure lags, with the most pronounced effects observed at lags 15 and 20 (Figure 2). Specifically, for lag 20, we observed an estimated RR of 1.19 (95% CI: 1.14; 1.23) for DI, 1.12 (95% CI: 1.08; 1.16) for H, 1.17 (95% CI: 1.12; 1.22) for HI, and 1.18 (95% CI: 1.14; 1.23) for NET (Figure 2). On the other hand, at high exposure (99th percentile), we observed that the shorter lags (3, 5, 7, and 10) had the most robust effects on all-cause mortality (Figure 3). For example, under lag 3, the pooled national results for the overall population (all ages and sexes) indicate an increased risk of all-cause mortality, with an RR of 1.17 (95% CI: 1.13; 1.122) for DI, 1.15 (95% CI: 1.12; 1.17) for H, 1.15 (95% CI: 1.09; 1.21) for HI, and 1.18 (95% CI: 1.13; 1.22) for NET (Figure 3).

For respiratory mortality, under low exposure levels (Figure 4), robust associations with thermal indices were observed primarily in certain sex/age groups, notably the overall population and elderly individuals, particularly at lags 15 and 20. NET was the only thermal index without a robust positive association. Considering the lag 20, exposures of the overall population (all ages and sexes) to low DI, H, and HI were significantly associated with an increased risk of respiratory mortality, with estimated RRs of 1.26 (95% CI: 1.10; 1.42), 1.26 (95% CI: 1.08; 1.44), and 1.25 (95% CI: 1.11; 1.39), respectively (Figure 4). In contrast, high exposure conditions (99th percentile) were linked to increased risk, particularly at shorter lags such as 3, 5, 7, and 10 days (Figure 5). For example, at lag 5, we observed RRs of 1.31 (95% CI: 1.12; 1.51) for DI, 1.22 (95% CI: 1.11; 1.34) for NET, 1.38 (95% CI: 1.09; 1.68) for H, and 1.31 (95% CI: 1.13; 1.48) for HI (Figure 5).

Finally, for circulatory mortality, low exposure conditions consistently showed robust positive associations across nearly all subgroup analyses, with the most pronounced effects observed at longer lags (Figure 6). For example, at lag 20, exposures of the overall population (all ages and sexes) to low DI, H, HI, and NET were significantly associated with an increased risk of circulatory mortality, with estimated RRs of 1.30 (95% CI: 1.18; 1.41), 1.32 (95% CI: 1.19; 1.44), 1.29 (95% CI: 1.18; 1.40), and 1.22 (95% CI: 1.13; 1.31), respectively (Figure 6). On the other hand, under high exposure conditions, robust positive associations were primarily observed at shorter lags (Figure 7). For example, at lag 5, high exposure levels of the overall population (all ages and sexes) to low DI, HI, and NET were significantly associated with an increased risk of circulatory mortality, with estimated RRs of 1.09 (95% CI: 1.02; 1.15), 1.07 (95% CI: 1.02; 1.11), and 1.06 (95% CI: 1.02; 1.10), respectively (Figure 7). Particularly, the thermal index H did not exhibit a robust positive association in these contexts.

## 4. Discussion

Our study suggests significant variations in the associations between thermal indices and mortality outcomes across different health categories, including all-cause mortality, respiratory mortality, and circulatory mortality. Specifically, we observed robust positive associations across nearly all subgroup analyses when considering all-cause mortality. This finding suggests a consistent and significant impact of thermal conditions on overall mortality rates, regardless of specific demographic or exposure factors. In contrast, for respiratory mortality, robust positive associations were predominantly observed in only a few subgroup analyses. This limited association may be attributed to the complex nature of respiratory health outcomes, influenced by factors such as underlying respiratory conditions, age, and individual susceptibility to thermal stress. Interestingly, circulatory mortality exhibited robust positive associations across almost all subgroup analyses when accounting for low exposure conditions, indicating a strong link between thermal stress and cardiovascular health outcomes across diverse demographic groups. However, under high exposure conditions, robust positive associations were observed in only a few subgroup analyses for circulatory mortality, suggesting a potential threshold effect where higher levels of thermal exposure may not uniformly impact circulatory mortality across all demographic groups.

These findings align with prior research indicating that different health outcomes may respond differently to thermal conditions [26,27] highlighting the complexity of the interactions between thermal conditions and health. From a biological point of view, these variations can be attributed to a range of interconnected biological mechanisms. First, physiological sensitivity plays a pivotal role, as the respiratory and circulatory systems exhibit varying degrees of susceptibility to thermal stressors [28,29,30]. Respiratory mortality may arise from direct thermal effects on respiratory function, exacerbating pre-existing conditions such as asthma and chronic obstructive pulmonary disease (COPD) [31,32]. In contrast, circulatory mortality may involve systemic responses to thermal stress, affecting cardiovascular function and vascular health [28,33,34]. Age-related vulnerabilities further contribute to these disparities, with older adults often exhibiting heightened susceptibility to thermal extremes due to compromised regulatory mechanisms [35,36,37]. Additionally, the presence of underlying health conditions, such as cardiovascular diseases or respiratory ailments, can modulate individual responses to thermal stress, amplifying mortality risks.

We also observe varying degrees of association depending on the lag period considered. For all-cause mortality, our results suggest consistent effects across various lag periods, irrespective of exposure levels, aligning with existing literature suggesting a broad impact of thermal conditions on overall mortality [38,39]. These findings underscore the comprehensive nature of thermal stress on human health, encompassing both short-lag and long-lag effects.

On the other hand, respiratory mortality exhibited a distinctive pattern, with robust positive associations predominantly observed at longer lag periods (specifically lag 15 and 20) under low exposure conditions, indicating a delayed response to thermal stress in respiratory health. This finding agrees with studies highlighting the cumulative impact of prolonged thermal exposure on respiratory function, potentially linked to inflammatory responses and respiratory system adaptation over time [30,40,41]. Conversely, under high exposure conditions for respiratory mortality, the effects were more pronounced at shorter lags (lags 3, 5, 7), reflecting an acute and immediate impact of extreme thermal conditions on respiratory health, supported by evidence suggesting rapid physiological responses to thermal stress, including airway constriction and increased susceptibility to respiratory infections [42,43].

Finally, for circulatory mortality, our findings indicate positive associations across most lag periods, particularly under low exposure conditions, indicating a sustained influence of thermal stress on cardiovascular health. This aligns with research indicating the role of thermal stress in exacerbating cardiovascular risk factors, such as increased blood pressure and heart rate variability, leading to adverse cardiovascular events [44]. Interestingly, under high exposure conditions for circulatory mortality, the positive associations extended to shorter lags (lags 3, 5, 7), suggesting an acute onset of cardiovascular effects in response to extreme thermal conditions [44]. One possible explanation for the different results (depending on the lag) between high and low exposure conditions for circulatory mortality lies in the acute physiological responses of the cardiovascular system to extreme thermal stress. High exposure conditions typically entail rapid and intense heat exposure, which can trigger immediate physiological reactions in the body [44]. For example, extreme heat can lead to vasodilation, where blood vessels expand to dissipate heat and maintain thermal equilibrium [45]. This process may occur very fast (in hours) and can result in increased blood flow and cardiac workload, potentially leading to acute cardiovascular events such as myocardial infarction or stroke [45]. Moreover, under high exposure conditions, individuals may experience more pronounced heat-related symptoms such as dehydration, electrolyte imbalance, and heat exhaustion, all of which can directly impact cardiovascular function [46].

Analyses by age showed differential susceptibility to thermal exposures. Elderly individuals consistently showed higher mortality risks compared to younger age groups. These findings align with existing literature suggesting age and sex as significant modifiers of the health impacts of environmental factors [35,36]. These observed variations can be elucidated through a range of biological, physiological, and behavioral factors. Elderly individuals, characterized by reduced physiological resilience, compromised thermoregulatory mechanisms, and a higher prevalence of chronic health conditions, face heightened risks during extreme thermal conditions [35,37]. Factors such as decreased cardiovascular reserve, impaired immune function, and age-related changes in thermoregulation contribute to their vulnerability, leading to elevated mortality risks [47]. Pre-existing health conditions like cardiovascular disease and respiratory disorders further exacerbate these risks, with heat stress potentially triggering acute cardiovascular events or respiratory complications [44]. Additionally, behavioral factors such as reduced mobility, inadequate hydration, and medication non-compliance during heat waves contribute to their susceptibility) [48].

We also found that the strength and direction of the associations varied among the indices. The DI, which combines average air temperature and relative humidity, consistently showed robust positive associations across different exposure scenarios. Its categorization of discomfort levels, including comfortable and uncomfortable ranges, provided valuable insights into the impact of thermal conditions on health outcomes. Similarly, the H, integrating air temperature and vapor pressure, exhibited strong positive associations, particularly in identifying mild to dangerous levels of thermal discomfort. In contrast, the NET, designed to evaluate outdoor thermal comfort comprehensively, showed mixed results, reflecting its sensitivity to diverse environmental conditions and its varied impact on mortality outcomes. Additionally, the HI, calculated based on air temperature and relative humidity, contributed insights into thermal discomfort levels, albeit with varying associations across different exposure scenarios. This comprehensive evaluation of multiple thermal indices underscores their utility in capturing the complex interplay between environmental conditions and health outcomes, emphasizing the importance of considering a range of indices for a holistic assessment of health risks related to thermal exposures.

While this study focuses exclusively on Brazil, a country with a unique combination of tropical and subtropical climates, the findings offer valuable insights into the relationship between thermal comfort and health outcomes. The socioeconomic disparities, geographic diversity, and climatic variability observed within Brazil make it an important case study. However, the results may not be directly generalizable to regions with different climates, such as temperate or arid zones, or to populations with distinct demographic and socioeconomic characteristics. Nevertheless, the use of thermal comfort indices such as the discomfort index, net effective temperature, humidex, and heat index provides a framework that can be adapted to other settings. These indices account for multiple environmental factors, which are universally relevant for assessing thermal stress. Future research could apply similar methodologies to regions with different climates to evaluate whether the patterns observed in Brazil hold true in other parts of the world. Additionally, studies could investigate how factors such as infrastructure, healthcare access, and local climate adaptation strategies influence the health impacts of thermal discomfort across diverse populations.

## 5. Conclusions

Our findings underscore the importance of considering multiple factors when assessing the health impacts of thermal exposures. Elderly individuals emerged as particularly susceptible to these exposures, aligning with existing literature on age and sex as significant modifiers of environmental health risks. Furthermore, the differential responses of health outcomes to thermal indices emphasize the need for nuanced approaches in public health interventions and policies, taking into account not only overall temperature but also specific indices and exposure durations.

In practical terms, these results highlight the necessity for targeted public health interventions, particularly in tropical regions like Brazil, where high temperatures and humidity can exacerbate health risks, especially among vulnerable populations such as the elderly. Public health strategies could include enhancing early warning systems that incorporate various thermal indices, improving urban planning to increase green spaces for cooling, and ensuring better access to air-conditioned environments during heat waves. Additionally, tailored communication campaigns could raise awareness about the specific risks associated with thermal discomfort, helping individuals and communities adopt protective behaviors during extreme weather events. These findings can inform not only national policies but also serve as a reference for countries with similar climates and socio-demographic profiles.

## Figures and Tables

**Figure 1 ijerph-21-01248-f001:**
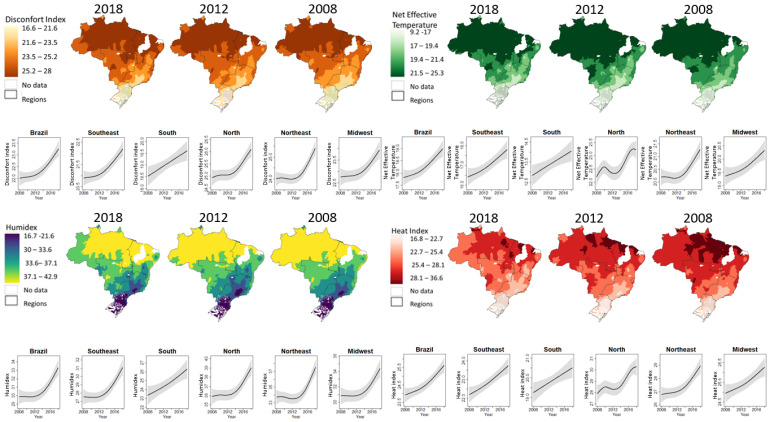
Spatiotemporal distribution of the thermal indices in Brazil.

**Figure 2 ijerph-21-01248-f002:**
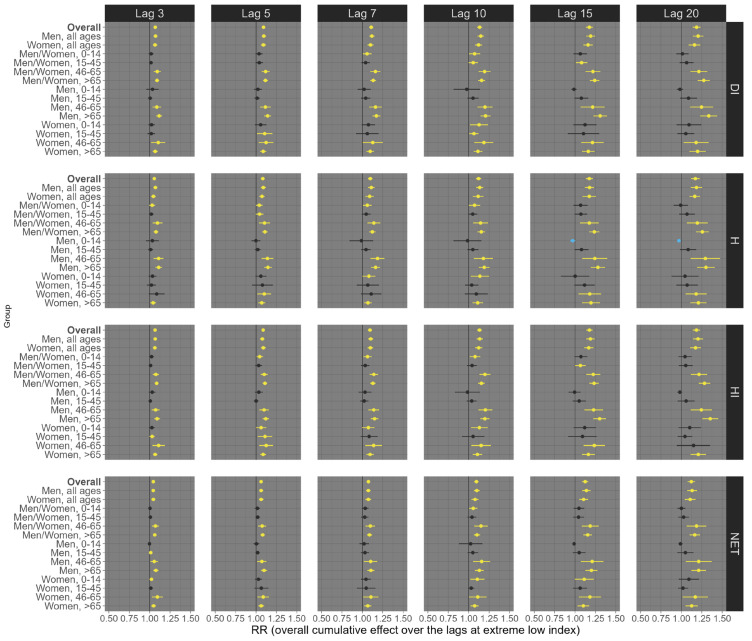
Relative risks (95%CI) of **all-cause mortality** for **low (1st percentile)** thermal index in Brazil (results from meta-analysis) stratified by age, sex, lag, and thermal index. Note 1: This is the overall cumulative effect over the lags (3, 5, 7, 10, 15, and 20), summing all the contributions up to the lag. Note 2: black color represents the insignificant coefficients (which the RR includes the value 1), yellow color represents the significant positive associations, and light blue color represents the significant negative associations.

**Figure 3 ijerph-21-01248-f003:**
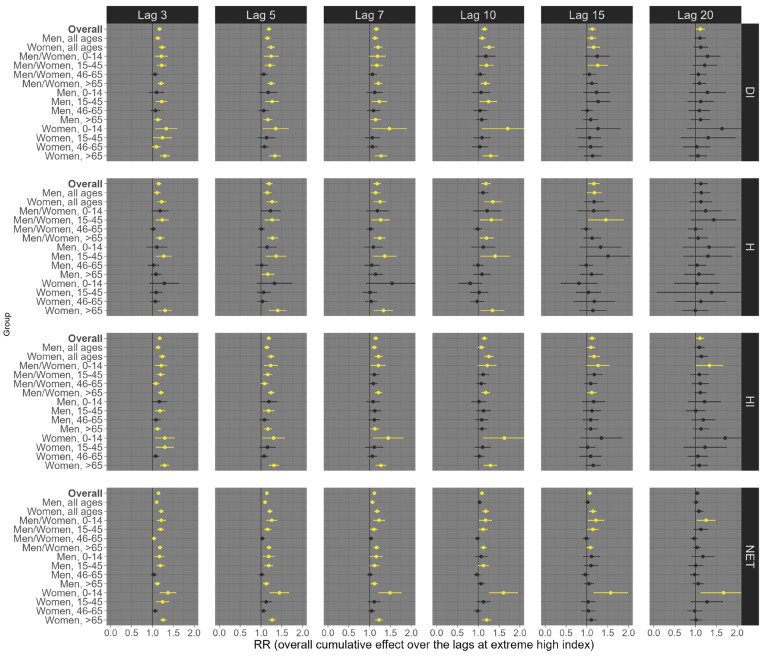
Relative risks (95%CI) of **all-cause mortality** for **high (99th percentile)** thermal index in Brazil (results from meta-analysis) stratified by age, sex, lag, and thermal index. Note 1: This is the overall cumulative effect over the lags (3, 5, 7, 10, 15, and 20), summing all the contributions up to the lag. Note 2: Black color represents the insignificant coefficients (which the RR includes the value 1), yellow color represents the significant positive associations.

**Figure 4 ijerph-21-01248-f004:**
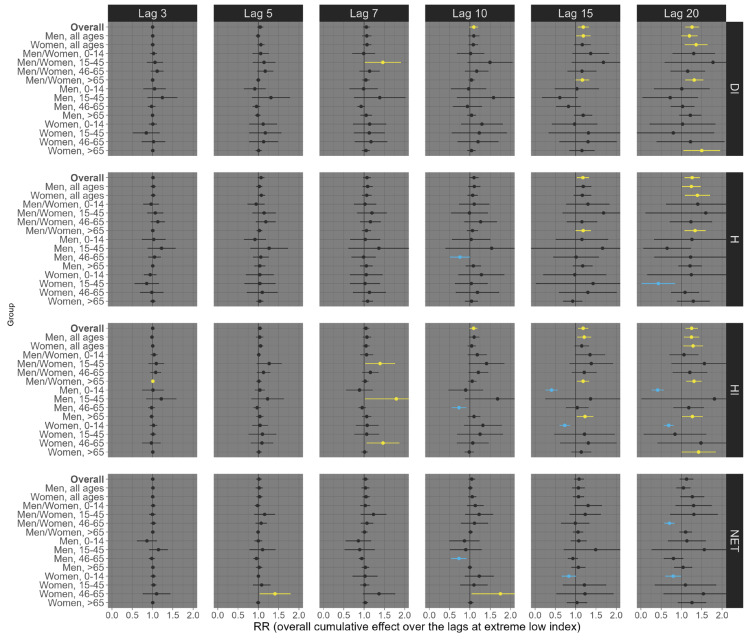
Relative risks (95%CI) of **respiratory mortality** for **low (1st percentile)** thermal index in Brazil (results from meta-analysis) stratified by age, sex, lag, and thermal index. Note 1: This is the overall cumulative effect over the lags (3, 5, 7, 10, 15, and 20), summing all the contributions up to the lag. Note 2: Black color represents the insignificant coefficients (which the RR includes the value 1), yellow color represents the significant positive associations, and light blue color represents the significant negative associations.

**Figure 5 ijerph-21-01248-f005:**
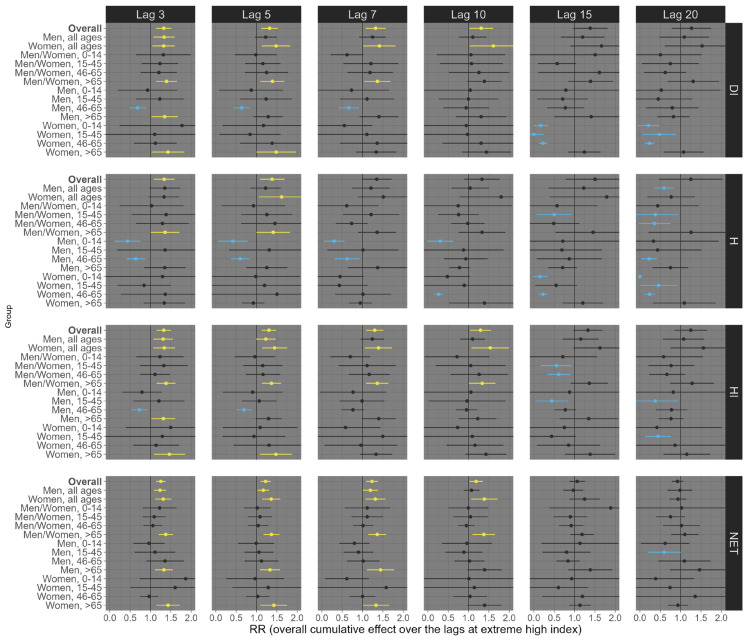
Relative risks (95%CI) of **respiratory mortality** for **high (99th percentile)** thermal index in Brazil (results from meta-analysis) stratified by age, sex, lag, and thermal index. Note 1: This is the overall cumulative effect over the lags (3, 5, 7, 10, 15, and 20), summing all the contributions up to the lag. Note 2: Black color represents the insignificant coefficients (which the RR includes the value 1), yellow color represents the significant positive associations, and light blue color represents the significant negative associations.

**Figure 6 ijerph-21-01248-f006:**
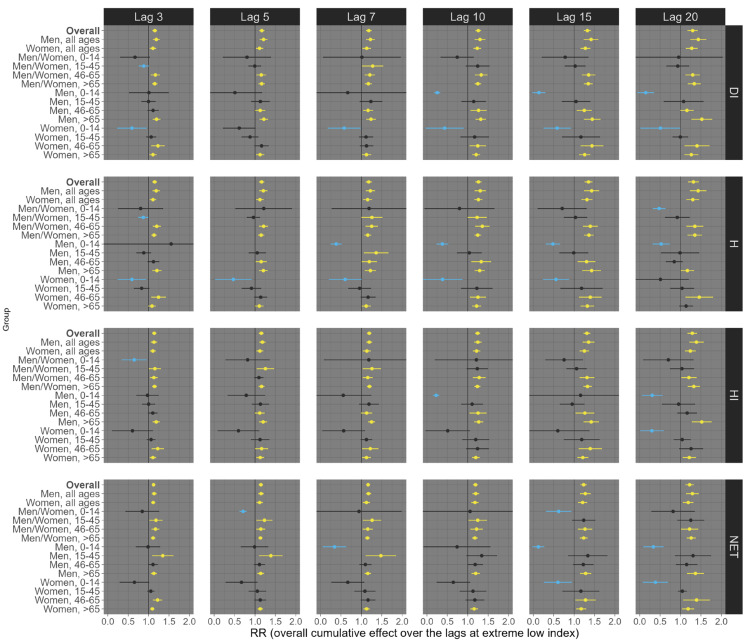
Relative risks (95%CI) of **circulatory mortality** for **low (1st percentile)** thermal index in Brazil (results from meta-analysis) stratified by age, sex, lag, and thermal index. Note 1: This is the overall cumulative effect over the lags (3, 5, 7, 10, 15, and 20), summing all the contributions up to the lag. Note 2: Black color represents the insignificant coefficients (which the RR includes the value 1), yellow color represents the significant positive associations, and light blue color represents the significant negative associations.

**Figure 7 ijerph-21-01248-f007:**
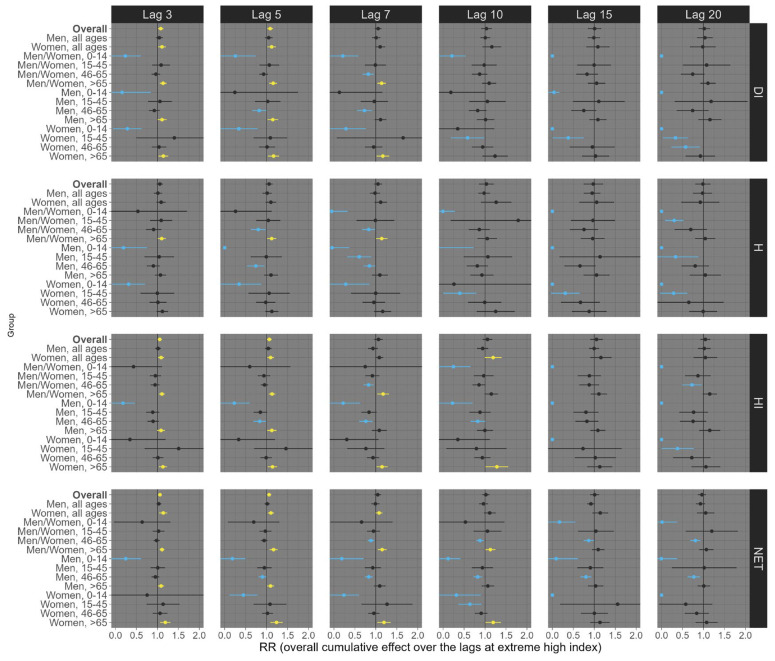
Relative risks (95%CI) of **circulatory mortality** for **high (99th percentile)** thermal index in Brazil (results from meta-analysis) stratified by age, sex, lag, and thermal index. Note 1: This is the overall cumulative effect over the lags (3, 5, 7, 10, 15, and 20), summing all the contributions up to the lag. Note 2: Black color represents the insignificant coefficients (which the RR includes the value 1), yellow color represents the significant positive associations, and light blue color represents the significant negative associations.

**Table 1 ijerph-21-01248-t001:** Demographic descriptions of mortality occurrences in Brazil from 2003 to 2017.

Health Outcome	Age	Number of Deaths (%) ^1^		
		Men	Women	All Sex
All-cause	0–14	123,948 (4.32)	94,750 (2.91)	218,698 (7.61)
	15–45	387,952 (13.51)	136,920 (0.39)	524,872 (18.27)
	46–65	444,165 (15.46)	270,087 (1.04)	714,252 (24.87)
	>65	734,314 (25.57)	722,730 (25.16)	1,457,044 (50.73)
	All ages	1,666,025 (58.01)	1,206,059 (41.99)	2,872,084 (100)
Respiratory	0–14	9554 (3.44)	7938 (2.53)	17,492 (6.29)
	15–45	13,511 (4.86)	8069 (0.32)	21,580 (7.77)
	46–65	30,212 (10.87)	19,407 (0.66)	49,619 (17.86)
	>65	100,198 (36.06)	92,878 (33.42)	193,076 (69.48)
	All ages	151,185 (54.41)	126,695 (45.59)	277,880 (100)
Circulatory	0–14	1852 (0.23)	1682 (0.13)	3534 (0.45)
	15–45	34,549 (4.36)	23,814 (0.08)	58,363 (7.36)
	46–65	134,814 (17)	87,699 (0.3)	222,513 (28.05)
	>65	256,733 (32.37)	268,236 (33.82)	524,969 (66.18)
	All ages	418,555 (52.77)	374,655 (47.23)	793,210 (100)

Note: ^1^ the percentages were based on the total number of deaths in Brazil between 2003 and 2017; there were 277,880 cases of mortality due to respiratory diseases; 793,210 cases due to circulatory diseases, and 2,872,084 cases of all-cause mortality.

**Table 2 ijerph-21-01248-t002:** Summary statistics for weather parameters and thermal indices in Brazil, 2008–2018.

Variable		Min	Q1	Mean	SD	Q3	Max
Weather parameters	Temperature (°C)	1.25	21.60	23.59	3.79	26.23	34.92
	Relative humidity (%)	20.00	69.75	77.76	13.64	88.50	100
	Wind speed (m/s)	0.10	2.05	2.99	1.25	3.72	17.62
Thermal indices	Discomfort index	2.39	20.71	22.41	3.24	24.74	31.95
	Net effective temperature	−12.17	15.85	18.26	4.59	21.64	31.49
	Humidex	−1.45	27.01	30.74	6.27	35.30	52.24
	Heat index	1.00	22.00	24.59	4.80	28.00	52.00

Note 1: Summary statistics were calculated considering the values at the municipality level averaged across the entire country in the study period. Note 2: Minimum (Min), first quartile (Q1), standard deviation (SD), third quartile (Q3), and maximum (Max).

## Data Availability

The original contributions presented in the study are included in the article/Appendix A, further inquiries can be directed to the corresponding author.

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
