# Peer review of "Thermal Comfort Conditions and Mortality in Brazil"

_ijerph, 2024, doi:10.3390/ijerph21091248_

Round 1

Reviewer 1 Report

Comments and Suggestions for Authors

The paper presents an interesting analysis of the association between thermal comfort conditions and mortality related to respiratory and circulatory diseases in Brazil. However, there are several areas where the study could be improved:

1) This introduction did not explain the research gaps. Current Introduction is not acceptable. Please add more thermal comfort related publications and discuss them.

2) Section 2, the authors do not report power calculations or discuss the statistical of deaths data.

3) Section 2.3, four thermal indices (DI, NET, H, and HI) are used to measure thermal comfort. This study does not discuss the strengths and limitations of each index or how they may differ in their ability to capture thermal stress.

4) Give calculation equations for four thermal indices if possible.

5) The study focuses exclusively on Brazil, which limits the generalizability of the findings to other regions with different climates and populations. The authors should discuss the potential applicability of their results to other settings.

6) Page 5, “a dataset of 2,872,084 recorded deaths in Brazil from 2003 to 2017”, give the reference or  data source.

7) Figures 2-7 are not shown in full.

8) The authors could strengthen the paper by discussing the practical implications of their findings for public health interventions in Conclusion section.

Author Response

The paper presents an interesting analysis of the association between thermal comfort conditions and mortality related to respiratory and circulatory diseases in Brazil. However, there are several areas where the study could be improved:

Response:

Thank you for your time and thoughtful review of our manuscript. We have carefully considered your suggestions to improve the manuscript.

1) This introduction did not explain the research gaps. Current Introduction is not acceptable. Please add more thermal comfort related publications and discuss them.

Response:

Thank you for your feedback. We appreciate your suggestion to expand the introduction to better highlight research gaps and include more publications related to thermal comfort. In response to your comment, we have revised the introduction to provide a more detailed discussion on the research gaps in the field of thermal comfort and health. We have also included additional relevant literature to frame the context of our study and underscore its importance. The revised introduction now elaborates on the existing gaps in the literature and integrates recent studies to provide a comprehensive overview of the current research landscape.

2) Section 2, the authors do not report power calculations or discuss the statistical of deaths data.

Response:

We appreciate the reviewer's insightful comment regarding the need to discuss power calculations and the statistical robustness of the mortality data. In response to this comment, we have included a paragraph in section 2.1 (lines 92-100).

“Although we did not perform a formal power calculation, the scale and comprehensiveness of the mortality data in this study inherently provide a strong basis for statistical power. The data were sourced from the Brazilian Ministry of Health’s Mortality Information System (SIM), the official and national repository of mortality records in Brazil. This system covers the entire country, ensuring a high degree of completeness and reliability in the data collection process. With over 1,071,090 deaths from circulatory and respiratory diseases included in the analysis, spanning 5,572 municipalities across 15 years, the size and national coverage of the dataset ensure a high likelihood of detecting meaningful associations. The large sample size, combined with a nationwide representation, increases the precision of our estimates and reduces the risk of Type II errors (failing to detect an effect when one exists).”

3) Section 2.3, four thermal indices (DI, NET, H, and HI) are used to measure thermal comfort. This study does not discuss the strengths and limitations of each index or how they may differ in their ability to capture thermal stress.

Response:

Thank you for the suggestion. We have added a discussion on the strengths and limitations of each index (lines 115-144):

  • “Discomfort Index (DI): This index merges average air temperature and relative humidity to gauge outdoor comfort levels. Values under 15 and above 26.5 signal discomfort, while those between 15 and 20 are considered comfortable, and values from 20 to 26.5 indicate slight discomfort. The DI combines air temperature and relative humidity to assess thermal comfort. One of its strengths lies in its simplicity and its long history of use in public health research. However, DI does not account for wind speed, which can be a critical factor in moderating heat stress, especially in warmer regions like Brazil. This limitation may reduce its ability to capture thermal discomfort in areas with high winds, where wind chill could affect perceived temperature.
  • Net Effective Temperature (NET): This index evaluates outdoor thermal comfort by factoring in air temperature, relative humidity, and wind speed. A NET value between 17 and 21 denotes comfort, with specific ranges indicating mild discomfort in colder conditions and discomfort in hotter conditions. Its ability to factor in wind speed makes it more accurate in representing outdoor thermal comfort in various climatic conditions, especially in tropical regions where wind can mitigate heat stress. However, NET is more complex to calculate, and its interpretation may be less intuitive than simpler indices like DI or HI, potentially limiting its broader use in public health studies.
  • Humidex (H): By combining air temperature and vapor pressure, the Humidex provides a measure of thermal discomfort. Values below 29 suggest no discomfort, those between 30 and 39 imply mild discomfort, and values exceeding 45 points to hazardous discomfort levels. The Humidex is well-suited to assess heat stress in regions with high humidity, as it reflects the combined impact of heat and moisture on human comfort. However, like DI, Humidex does not consider wind speed, which may lead to an incomplete assessment of thermal stress in windy environments. Additionally, Humidex values can sometimes overestimate discomfort in regions with extremely high humidity levels, which may exaggerate the associated health risks.
  • Heat Index (HI): This index, derived from air temperature and relative humidity, indicates levels of thermal discomfort. Values under 27 are deemed comfortable, while those over 32 are classified as dangerous. One of HI's strengths is its familiarity and extensive use in heat-health warning systems, which makes it practical for public health applications. However, similar to DI and Humidex, HI does not include wind speed, which may limit its applicability in windy or cooler regions where wind plays a significant role in moderating thermal discomfort.

For comprehensive information on the formulation and interpretation of each index, please refer to the supplemental materials. Overall, the inclusion of multiple indices in this study allows us to capture different aspects of thermal stress across a range of climatic conditions. While indices like NET offer a more comprehensive measure by incorporating wind speed, simpler indices such as DI, Humidex, and HI are easier to calculate and interpret. Each index has strengths in different settings, but limitations arise when key meteorological factors—such as wind speed—are excluded, potentially leading to under- or overestimation of thermal stress in specific environments. By using a combination of these indices, we aim to provide a more nuanced understanding of the relationship between thermal comfort and health outcomes in Brazil’s diverse climatic regions.”

4) Give calculation equations for four thermal indices if possible.

Response:

Thank you for your comment. The calculation equations for the four thermal indices are included in the supplementary materials. We have referenced this in the manuscript (Section 2.3)

5) The study focuses exclusively on Brazil, which limits the generalizability of the findings to other regions with different climates and populations. The authors should discuss the potential applicability of their results to other settings.

Response:

Thank you for the suggestion. We have added a paragraph in the discussion section that acknowledges the study's regional focus while positioning the findings as a starting point for broader research. It also highlights the potential adaptability of the methodology to other contexts (lines 346-357):

While this study focuses exclusively on Brazil, a country with a unique combination of tropical and subtropical climates, the findings offer valuable insights into the relationship between thermal comfort and health outcomes. The socioeconomic disparities, geographic diversity, and climatic variability observed within Brazil make it an important case study. However, the results may not be directly generalizable to regions with different climates, such as temperate or arid zones, or to populations with distinct demographic and socioeconomic characteristics. Nevertheless, the use of thermal comfort indices such as the Discomfort Index, Net Effective Temperature, Humidex, and Heat Index provides a framework that can be adapted to other settings. These indices account for multiple environmental factors, which are universally relevant for assessing thermal stress. Future research could apply similar methodologies to regions with different climates to evaluate whether the patterns observed in Brazil hold true in other parts of the world. Additionally, studies could investigate how factors such as infrastructure, healthcare access, and local climate adaptation strategies influence the health impacts of thermal discomfort across diverse populations.”

6) Page 5, “a dataset of 2,872,084 recorded deaths in Brazil from 2003 to 2017”, give the reference or  data source.

Response:

Thank you. We have added the data source – “Brazilian Ministry of Health”.

7) Figures 2-7 are not shown in full.

Response:

Thank you for pointing out the issue with Figures 2-7. I believe there was an error during the submission process when the manuscript file was uploaded and converted by the journal's system. The figures are complete and appear correctly in the original file. I will ensure that the figures are properly displayed in the revised submission. As reference, I show below the figure 2-7:

Figure 2 - Relative risks (95%CI) of all-cause mortality for low (1st percentile) thermal index in Brazil (results from meta-analysis) stratified by age, sex, lag, and thermal index. Note 1: This is the overall cumulative effect over the lags (3, 5, 7, 10, 15, and 20), summing all the contributions up to the lag. Note 2: dark blue color represents the insignificant coefficients (which the RR includes the value 1), yellow color represents the significant positive associations, and light blue color represents the significant negative associations.

Figure 3 - Relative risks (95%CI) of all-cause mortality for high (99th percentile) thermal index in Brazil (results from meta-analysis) stratified by age, sex, lag, and thermal index. Note 1: This is the overall cumulative effect over the lags (3, 5, 7, 10, 15, and 20), summing all the contributions up to the lag. Note 2: dark blue color represents the insignificant coefficients (which the RR includes the value 1), yellow color represents the significant positive associations, and light blue color represents the significant negative associations.

Figure 4 - Relative risks (95%CI) of respiratory mortality for low (1st percentile) thermal index in Brazil (results from meta-analysis) stratified by age, sex, lag, and thermal index. Note 1: This is the overall cumulative effect over the lags (3, 5, 7, 10, 15, and 20), summing all the contributions up to the lag. Note 2: dark blue color represents the insignificant coefficients (which the RR includes the value 1), yellow color represents the significant positive associations, and light blue color represents the significant negative associations.

Figure 5 - Relative risks (95%CI) of respiratory mortality for high (99th percentile) thermal index in Brazil (results from meta-analysis) stratified by age, sex, lag, and thermal index. Note 1: This is the overall cumulative effect over the lags (3, 5, 7, 10, 15, and 20), summing all the contributions up to the lag. Note 2: dark blue color represents the insignificant coefficients (which the RR includes the value 1), yellow color represents the significant positive associations, and light blue color represents the significant negative associations.

Figure 6 - Relative risks (95%CI) of circulatory mortality for low (1st percentile) thermal index in Brazil (results from meta-analysis) stratified by age, sex, lag, and thermal index. Note 1: This is the overall cumulative effect over the lags (3, 5, 7, 10, 15, and 20), summing all the contributions up to the lag. Note 2: dark blue color represents the insignificant coefficients (which the RR includes the value 1), yellow color represents the significant positive associations, and light blue color represents the significant negative associations.

Figure 7 - Relative risks (95%CI) of circulatory mortality for high (99th percentile) thermal index in Brazil (results from meta-analysis) stratified by age, sex, lag, and thermal index. Note 1: This is the overall cumulative effect over the lags (3, 5, 7, 10, 15, and 20), summing all the contributions up to the lag. Note 2: dark blue color represents the insignificant coefficients (which the RR includes the value 1), yellow color represents the significant positive associations, and light blue color represents the significant negative associations.

8) The authors could strengthen the paper by discussing the practical implications of their findings for public health interventions in Conclusion section.

Response:

Thank you for the suggestion. We have added this discussion (lines 367-375):

“In practical terms, these results highlight the necessity for targeted public health interventions, particularly in tropical regions like Brazil, where high temperatures and humidity can exacerbate health risks, especially among vulnerable populations such as the elderly. Public health strategies could include enhancing early warning systems that incorporate various thermal indices, improving urban planning to increase green spaces for cooling, and ensuring better access to air-conditioned environments during heatwaves. Additionally, tailored communication campaigns could raise awareness about the specific risks associated with thermal discomfort, helping individuals and communities adopt protective behaviors during extreme weather events. These findings can inform not only national policies but also serve as a reference for countries with similar climates and socio-demographic profiles.”

Reviewer 2 Report

Comments and Suggestions for Authors

The aim of the article entitled "Thermal comfort conditions and mortality in Brazil" was to determine the relationship between thermal comfort conditions and mortality related to respiratory and circulatory diseases. The authors examined four distinct thermal indices: the Discomfort 12 Index (DI), Net Effective Temperature (NET), Humidex (H), and Heat Index (HI).

Considering the ongoing climate changes and the need to take action to adapt the population to the ongoing changes, the aim of the article is fully justified. The authors of the article used modern research methods to achieve the assumed research goal. The work was based on very well-selected literature on the subject of research. The obtained results are interesting and deepen the knowledge of the impact of meteorological conditions on mortality related to respiratory and circulatory diseases. The conducted analyses of the impact of the delay in reaction to atmospheric conditions are worth noting. I rate the article very good, however, the authors did not avoid several minor shortcomings.

Minor remarks:
1. The article should be accompanied by a map of the research area (Brazil) against the background of the South American continent and a map of Brazil divided into regions included in the article.
2. Figure 1 contains errors. The graphs show long-term temperature trends, not the analyzed indicators. 3. Figures 6 and 7 do not fit on the pages.

Author Response

The aim of the article entitled "Thermal comfort conditions and mortality in Brazil" was to determine the relationship between thermal comfort conditions and mortality related to respiratory and circulatory diseases. The authors examined four distinct thermal indices: the Discomfort 12 Index (DI), Net Effective Temperature (NET), Humidex (H), and Heat Index (HI).

Considering the ongoing climate changes and the need to take action to adapt the population to the ongoing changes, the aim of the article is fully justified. The authors of the article used modern research methods to achieve the assumed research goal. The work was based on very well-selected literature on the subject of research. The obtained results are interesting and deepen the knowledge of the impact of meteorological conditions on mortality related to respiratory and circulatory diseases. The conducted analyses of the impact of the delay in reaction to atmospheric conditions are worth noting. I rate the article very good, however, the authors did not avoid several minor shortcomings.

Response:

Thank you for your thoughtful review and for recognizing the relevance of our study. We appreciate your feedback, and we have carefully addressed each of your concerns in the revised manuscript. We have outlined major revisions as per your suggestions and believe these will strengthen the overall quality of the work.

Minor remarks:

  1. The article should be accompanied by a map of the research area (Brazil) against the background of the South American continent and a map of Brazil divided into regions included in the article.

Response:

Thank you for the suggestion. I would like to clarify that the map of the research area (Brazil), including its division into regions, is already presented in Figure S1, which is included in the supplementary materials. This is mentioned in Section 2.1, at the end of the first paragraph. Given the already high number of figures included in the main manuscript (7 figures), I believe it is appropriate to keep this figure in the supplementary materials to maintain the overall balance of the paper.

  1. Figure 1 contains errors. The graphs show long-term temperature trends, not the analyzed indicators.

Response:

Thank you. We have fixed it.

  1. Figures 6 and 7 do not fit on the pages.

Response:

Thank you for pointing out the issue with Figures 6-7. I believe there was an error during the submission process when the manuscript file was uploaded and converted by the journal's system. The figures are complete and appear correctly in the original file. I will ensure that the figures are properly displayed in the revised submission.

Round 2

Reviewer 1 Report

Comments and Suggestions for Authors

Authors address my concerns.